# Identification and characterization of tweets related to the 2015 Indiana HIV outbreak: A retrospective infoveillance study

**Mingxiang Cai[1,2,3], Neal Shah[1,2], Jiawei Li[1,2,4], Wen-Hao Chen[2,3], Raphael E. Cuomo[1,5], Nick Obradovich[6], Tim K. Mackey**[1,2,5,7] *

**1** Global Health Policy Institute, San Diego, CA, United States of America, **2** Department of Healthcare Research and Policy, University of California, San Diego, CA, United States of America, **3** Department of Computer Science and Engineering, University of California, San Diego, CA, United States of America, **4** Department of Computational Science, Mathematics and Engineering, University of California, San Diego, CA, United States of America, **5** Department of Anesthesiology, San Diego School of Medicine, University of California, San Diego, CA, United States of America, **6** Max Planck Institute for Human Development, Berlin, Germany, **7** Division of Infections Disease and Global Public Health, Department of Medicine, San Diego School of Medicine, University of California, San Diego, CA, United States of America

* tmackey@ucsd.edu

**Data Availability Statement:** An anonymized Twitter data set containing our target dataset is available on first author's GitHub repository at: https://github.com/marcopolocai/Identification-

## Abstract

### Introduction

From late 2014 through 2015, Scott County, Indiana faced an HIV outbreak triggered by opioid abuse and transition to injection drug use. Investigating the origins, risk factors, and responses related to this outbreak is critical to inform future surveillance, interventions, and policymaking. In response, this retrospective infoveillance study identifies and characterizes user-generated messages related to opioid abuse, heroin injection drug use, and HIV status using natural language processing (NLP) among Twitter users in Indiana during the period of this HIV outbreak.

### Materials and methods

Our study consisted of two phases: data collection and processing, and data analysis. We collected Indiana geolocated tweets from the public Twitter API using Amazon Web Services EC2 instances filtered for geocoded messages in the immediate pre and post period of the outbreak. In the data analysis phase we applied an unsupervised machine learning approach using NLP called the Biterm Topic Model (BTM) to identify tweets related to opioid, heroin/injection, and HIV behavior and then examined these messages for HIV risk-related topics that could be associated with the outbreak.

### Results

More than 10 million geocoded tweets occurring in Indiana during the immediate pre and post period of the outbreak were collected for analysis. Using BTM, we identified 1350 tweets thought to be relevant to the outbreak and then confirmed 358 tweets using human annotation. The most prevalent themes identified were tweets related to self-reported abuse

and-Characterization-of-Tweets-related-to-the-
2015-Indiana-HIV-Outbreak-Using-ML

**Funding:** Authors received an AIDS Research
Development Grant from the UC San Diego Center
for AIDS Research (CFAR) (NIAD 5 P30 A1036214)
in support of this study.

**Competing interests:** The authors have declared
that no competing interests exist.

of illicit and prescription drugs, opioid use disorder, self-reported HIV status, and public sentiment regarding the outbreak. Geospatial analysis found that these messages clustered in population dense areas outside of the outbreak, including Indianapolis and neighboring Clark County.

## Discussion

This infoveillance study characterized the social media conversations of communities in Indiana in the pre and post period of the 2015 HIV outbreak. Behavioral themes detected reflect discussion about risk factors related to HIV transmission stemming from opioid and heroin abuse for priority populations, and also help identify community attitudes that could have motivated or detracted the use of HIV prevention methods, along with helping identify factors that can impede access to prevention services.

## Conclusions

Infoveillance approaches, such as the analysis conducted in this study, represent a possibly strategy to detect "signal" of the emergence of risk factors associated with an outbreak though may be limited in their scope and generalizability. Our results, in conjunction with other forms of public health surveillance, can leverage the growing ubiquity of social media platforms to better detect opioid-related HIV risk knowledge, attitudes and behavior, as well as inform future prevention efforts.

## Introduction

In December 2014, public health officials at the Indiana State Department of Health started to notice a concerning trend–HIV cases in and around Scott County, the rural, southeastern part of Indiana, rose at a startling rate [1]. By April of 2015, new HIV infections had officially reached alarming numbers with a total of 135 confirmed HIV infections [1]. The emergence of this localized rural HIV outbreak highlighted existing concerns already voiced by the public health community about the dangerous connection between prescription opioid abuse, heroin injection drug use (IDU), lack of access to needle exchange programs (NEP), and related HIV risk [2,3]. The subsequent 169 cases (representing a 3000% increase from prior annual averages) reported in the first half of 2015, marked a sentinel event in understanding the behavioral transition between prescription opioids and heroin-injection drug use leading to HIV transmission [2]. Hence, it is necessary to explore the origins of this outbreak to better understand how they contribute to the HIV risk environment and prevention cascades and how lessons learned can contribute to future targeted HIV prevention strategies [4,5].

One avenue of investigation is using large volumes of unstructured data to identify and characterize infectious diseases in the context of data-informed surveillance, also known as infoveillance [6]. This specifically includes using "big data" generated by now ubiquitous social media platforms that can supplement structured data gathered via laboratory confirmed tests or national population-based surveys that measure prevalence estimates, attitudes, and associated trends of various forms of substance abuse and HIV behavior [7,8]. The popular microblogging social media platform Twitter has previously been used as a source of user-generated data to explore many infectious disease-related topics including influenza, misinformation about vaccination, the Ebola outbreak, and pre-exposure prophylaxis [9–12]. Within HIV

specifically, social media has been used to identify safe and risky sexual behaviors associated with vulnerable populations, better integrate harm reduction services, and create a personalized risk assessment tool for high-risk groups [13–17]. Social media has also demonstrated utility specifically with HIV prevention efforts [18].

In addition, other studies have specifically examined the 2015 Indiana HIV outbreak focusing on uncovering its root causes, specific strategies that could have prevented the outbreak, and modeling whether earlier responses to the outbreak could have had an impact on the number of people infected [2,3,19,20]. Building on this prior research, we conducted a retrospective infoveillance study with an unsupervised machine learning approach using natural language processing (NLP) to identify and characterize messages on Twitter posted in the immediate pre and post period of the Indiana outbreak.

## Materials and methods

The aims of this study focused on using big data and NLP to detect, classify and characterize Twitter messages associated with opioid abuse, heroin injection drug use, and HIV user behaviors and attitudes. To carry out these aims, the study was conducted in 2 distinct phases: data collection and processing, and data analysis, which we describe below (see Fig 1 for summary of methods). All data collection and analyses were done in the computer programming language Python and related Python packages. The study was deemed exempt and not qualified as human subjects research and the need for consent was waived by the UC San Diego Human Research Protections Program Institutional Review Board (Project #170151XX).

### Data collection and processing

Twitter allows access to publicly posted tweets through a public streaming application programming interface (API) per the terms of its Developer Agreement and Policies. We used Amazon Web Services (AWS) cloud-based computing services to collect tweets filtered for geocoded messages (user-enabled sharing of location coordinates) from the Twitter API from January 2014 to March 2016. Within this dataset, tweets from October 2014 to June 2015 (the period 6 months prior and 3 months after the outbreak was widely reported, though we note that the US Centers for Disease Control and Prevention reports that the outbreak started as early as 2011) were isolated and served as our dataset used for analysis in this study. This time limited dataset consisted of 585,620,471 tweets geocoded for different locations in the United States; each tweet contained the text content of the tweet and additional metadata such as user information, the time stamp, retweet status, and self-reported/detected geolocation (i.e. longitude and latitude coordinates).

After isolating tweets timestamped during the outbreak period, we filtered tweets for a geospatial shape file limited to the state of Indiana that resulted in a total of 10,148,663 tweets. We then applied a text filtering process using keywords based on three topics: HIV, heroin, and opioids similar to keywords used in other related studies. We also filtered our dataset for common "street" or "slang" terms associated with heroin and IDU and prescription opioid drug abuse (see Table 1 for a list of filtered keywords). This provided us with a target dataset of 5,112 tweets consisting of messages geolocated in Indiana and that also contained a relevant targeted keyword that we then processed for thematic analysis using NLP.

An anonymized data set containing our target dataset is available on first author's GitHub repository at: https://github.com/marcopolocai/Identification-and-Characterization-of-Tweets-related-to-the-2015-Indiana-HIV-Outbreak-Using-ML.

**Fig 1. Summary of data collection and processing methodology.**

**Table 1. Keywords used for filtering.**

| Category: | Keywords: |
|---|---|
| HIV | HIV, AIDS, HIV/AIDS |
| Heroin | Heroin, smack, speedball, screwball, tar, black tar, skag, china white, chiva, injection, inject, shoot up, shooting up, give wings, main line, slam, spike up. |
| Opioids | Hillbilly heroin, oxy, oxycotton, percs, happy pills, vikes, captain cody, sizzurp, purple drank, doors & floors, china white, goodfella, Murder 8, tango and cash, watson-387, dillies, smack, demmies, amidone, fizzies, Miss Emma, oxycet, blue heaven, Mrs. O, O bomb, oxy 80s, rushbo, morph, octagons, hydros |

## Data analysis

In order to achieve maximum automation of the data filtering process for study-related tweets, the application of machine learning and natural language processing (NLP) is an effective strategy to characterize messages that are unstructured and "free" (i.e. that have not been curated or labelled). Since we did not have a training set to classify messages related to our study aims, we employed an unsupervised machine learning approach designed to detect patterns in the data and summarize the content of the entire tweet corpus into distinct topics. Furthermore, this allowed us to identify and exclude non-behavioral tweets (e.g. news reports, bot traffic, advertisements, etc.) from our final results dataset. These topic modeling strategies are particularly suited for sorting short text into prevalent themes. We utilized the Biterm Topic Model (BTM), that identifies patterns in short texts and that has been used in prior studies examining substance abuse-related topics [14,21].

BTM is a topic clustering method that generates similar text into the same set of topics. Before using BTM, data was cleaned with only the text content from each tweet extracted, with links, numbers, special characters, and stop words removed. The resulting text for each filtered keyword was then categorized into topic clusters using BTM. We split all text into a bag of words and then used BTM to produce a discrete probability distribution for all words for each theme with the distribution placing a larger weight on words that are most representative of a given theme [14] Conducting BTM analysis is done by initially setting the BTM topic number (k) and "n" words (for the first round of analysis we set to k = 10, n = 20 to cover several possible topics.)

Based on the BTM output, we identified messages with "signal" characteristics if they included:

a. HIV, injection, or opioid-related keyword and a term and/or slang term associated with behavior;

b. Did not contain a term or set of terms associated with news media (e.g. "aids n", "am-news", @fox19").

The above characteristics were chosen primarily to remove news and non-user generated tweets from our filtered dataset. This process allowed us to use BTM to filter out thousands of tweets unrelated to the study aims (i.e. here we define "signal" as tweets related to HIV, opioid, or heroin/IDU self-reported user knowledge, attitudes or behavior), and then isolate "signal" tweets with specific relevance to our study. Due to BTM's topic modeling technique, news tweets and behavioral tweets tend to be assigned to different topic clusters, so we were able to isolate the news clusters and irrelevant behavioral clusters and excluded them from further analysis.

A subset of context-level topics which we believed were relevant to HIV, opioids, or injection drug use behavior were then extracted and analyzed qualitatively by human coders to characterize specific messages. Two human coders (first and second author), informed by a code book designed for this study, manually annotated the filtered subset of tweets generated from BTM themes. First and second author coded posts independently and achieved high intercoder reliability for results (kappa = 0.94). For inconsistent results, both the authors met and reviewed the posts together with the last author and decided on the correct classification of the post.

When manually annotating tweets detected after BTM, we used a coding protocol that focused on isolating tweets originating from human users discussing knowledge, attitudes and behaviors on the three topics, and excluding tweets that were news, other media content or non-drug uses of phrases (i.e. when a drug slang term was used for non-drug related

messages). We also removed retweets of news or media articles that originated from individual users. Though the content of news articles can be relevant to public perception and sentiment regarding the outbreak, they do not describe self-reported user behavior or direct observations of second-hand behavior, and hence were excluded. We then iteratively coded the tweets for prevailing themes relevant to the three major topics of HIV, injection drug use, and opioids. Sub-themes detected in these parent categories included: diversion, prevention, usage, treatment, specific mention of drug use, public opinion and sentiment, and political reaction.

The coders specifically looked for messages related to macro substance abuse and HIV behavioral themes and marked them relevant if they included self-reported opioid abuse behavior or product sourcing, self-reported injection drug use or mention of heroin use, self-report of HIV-risk related behaviors (e.g. unprotected sex, mention of multiple partners, men who have sex with men, etc.) or HIV status, and self-reporting of polydrug abuse. Additionally, coders searched for behavioral tweets centered around HIV awareness, prevention, misinformation, public sentiment, and the political reaction attributable to the outbreak. These themes were examined to assess aforementioned community attitudes towards barriers to access for critical prevention services, such as HIV testing centers and syringe exchange programs.

## Results

From the initial 10 million geocoded tweets available for this study, our data filtering and unsupervised machine learning methodology using BTM yielded a total of 1,350 tweets from Indiana that included keywords related to HIV (n = 487), heroin (n = 592), injection drug use (271), and opioids (505), detected during the pre and post period of the 2015 HIV outbreak that we believed to be user-generated signal tweets. Human annotation of this filtered dataset confirmed a total of 358 tweets made by individual users from Indiana that were deemed relevant to the 2015 Indiana HIV outbreak. Of these 358 tweets, 41.9% (n = 150) were associated with HIV, 37.2% (n = 133) were associated with heroin IDU, and 20.9% (n = 75) were associated with opioid use disorder. In total, 32.7% (n = 117) of the relevant tweets in the dataset were in the pre period of the outbreak, while 67.3% (n = 241) tweets were in the post period (see Table 2 for summary).

The most prevalent themes found in the entire "signal" dataset related to abuse of illicit and prescription drugs; a total of 133 tweets were identified as related to heroin IDU, and 75 tweets were related to opioid use or abuse (see Fig 2). The most common sub-themes themes included in our dataset included specific discussion of types of opioid use, mention of heroin injection behavior, reporting of HIV status, and public sentiment discussion following the outbreak. Table 3 provides examples of tweets from each thematic category that are also described below.

Tweets related to heroin, illicit drug use, or prescription opioid abuse included self-reporting by users of their own substance abuse behavior, documenting or alleging substance use by other users, self-reporting future heroin or prescription drug abuse, and commenting about

**Table 2. Frequency of relevant tweets in dataset per category after BTM.**

|  | HIV | Heroin/IDU | Opioids | Sum |
|---|---|---|---|---|
| Total Target Tweets | 487 | 863 | 505 | 1350 |
| Relevant | 150 | 133 | 75 | 358 |
| Relevant (%) in category | 30.8% | 15.4% | 14.9% | 26.5% |
| Pre outbreak (%) | 39.3% | 25.6% | 32% | 32.7% |
| Post outbreak (%) | 60.7% | 74.4% | 68% | 67.2% |

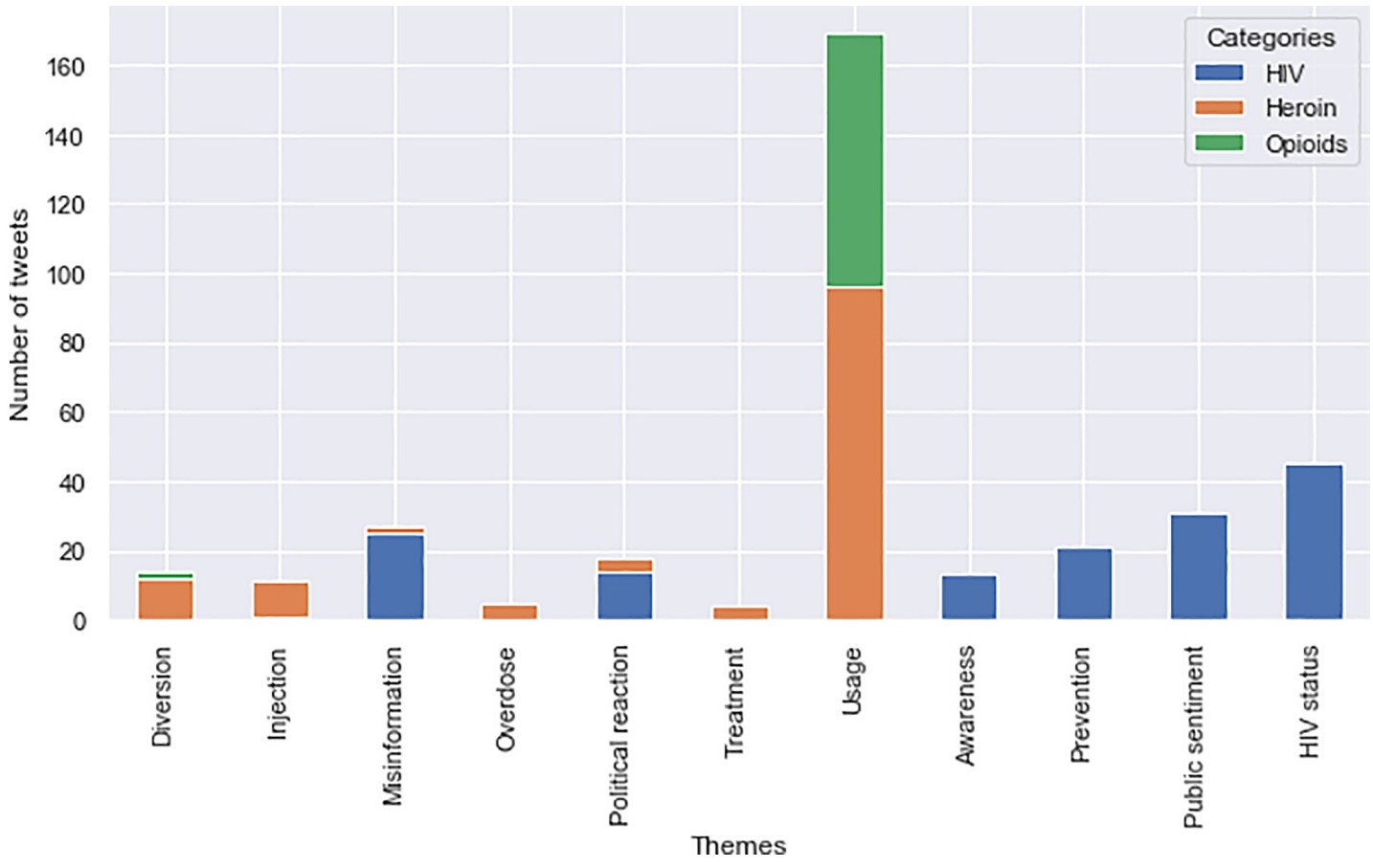

**Fig 2. Thematic organization of relevant tweets.**

substance use behavior in the context of public reactions to the outbreak. Additionally, 14 tweets about drug diversion were found in the dataset, along with 5 tweets about heroin-related overdose, and 11 tweets specifically reported injection of heroin by users. Importantly, tweets related specifically to injection reported explicit instances of self-injection drug use, finding needles/syringes, and witnessing injection drug use by other users, all key risk factors related to the outbreak [3].

From the HIV keyword group, a total of 45 tweets were collected that were related to HIV status. These tweets included self-reporting of purported user HIV status, documenting others' HIV status, and making disparaging/stigmatizing remarks about HIV status in the community. Misinformation was also detected; 25 tweets included incorrect and misleading statements about HIV risk and transmission. These messages included explicit statements mischaracterization the origin of the HIV outbreak.

Public sentiment and political reaction specific to the circumstances of the outbreak were also detected in the post-outbreak period. Specifically, political reactions included criticisms of Indiana's lack of NEPs, coupled with discussion about defunding of Planned Parenthood, which provided HIV prevention services in the State. Furthermore, a total of 13 tweets were about HIV awareness–including educational programs and events around reducing HIV transmission risks and accessing prevention resources. After the outbreak 17 tweets about HIV prevention (discussions about HIV testing, documenting use of pre-exposure prophylaxis

**Table 3. Examples of relevant tweets.**

Heroin:
  1. People in garrett are mixing heroin and meth and shooting it up. . .upper and a downer . . .. dumb (Injection)
  2. Literally a guy asked me for weed as I walked in the door. And then for heroin as I left. (Usage)
  3. apparently heroin is the new cute to do lmao (Usage)
  4. a kid who used to hit me up almost daily got arrested for dealing heroin last week. . .glad I attract scumbags (Diversion)
  5. I'm gnoing home and ovredsoing on heroin and cOCaine (Overdose)
  6. Such a fucked up world anymore I can get heroin 10 times easier than weed (Diversion)
  7. Lol people were doing Heroin at Bridge 11. Never again (Usage)

Heroin (street/slang):
  1. In need of a skag (Usage)
  2. she like where the tar at (Usage)
  3. blaq tar herion (Usage)

HIV (pre-outbreak):
  1. That's why y'all dick got EBOLA n HIV always tryna find out bout some pussy (Status)
  2. Straight just got aids (Status)
  3. Eeeeeewwwwww she said on my period AND I have HIV EWWWWWW (Status)
  4. Today in class I talked about how the government spread HIV and other disease around and no one believes me (Misinformation)
  5. Can the CIA give us the cure for AIDS before it drops Ebola off for the weekend? (Misinformation)

HIV (post-outbreak)
  1. this HIV outbreak came from sharing needles lol, yall making it more than what it is. . . . yall having sex with crackheads? (Public sentiment)
  2. Kids in FW are shooting pain killers with needles and sharing them and spreading HIV all over FW. (Usage)
  3. HIV outbreak in Scottsburg, IN and the idiotic 'religious freedom' bill are bringing Indiana into a wonderful spotlight. Way to go, guys. (Political reaction)
  4. Instead of catering to religious gps, Indiana government should focus on schools/roads, attracting business, meth and HIV epidemics, etc. (Political reaction)
  5. My legs been glued shut ever since i heard about that HIV shit in Indiana = it's not a game (Prevention)
  6. Long ago and far away I thought needle exchanges were complicit in addiction. Then a 72yr old AA minister schooled us at a convention. #HIV (Awareness)
  7. Indiana has an HIV positive outbreak! Those who are having sex, please be sure your parter is CLEAN & be safe (Prevention)

Opioids
  1. Who's trying to buy these hydros tho (Diversion)
  2. High off these percs (Usage)
  3. bout to take an oxy (Usage)
  4. Got couple Percs left who need em (Diversion)
  5. Got like 5 bottles of percs in my closet. . .ima a pharmacist (Diversion)
  6. Doug came in clutch with that Oxy tho. (Usage)
  7. thanksgiving in my family-dad: you don't have any hydros do you? aunt: yeah (Diversion)
  8. Opana 😡 (Usage)

(PrEP), encouraging or reporting safe sex habits) were also identified, with only 4 HIV prevention related tweets detected before the outbreak.

In addition to the manual annotation of tweets for thematic detection already described, we also visualized all relevant labelled signal tweets to exact longitude and latitude geocoordinates (n = 260) on a geospatial map of the State of Indiana (see Fig 3) using ArcGIS Map. Based on the geographic distribution of this data, the majority of tweets were located in areas of higher population density (e.g. Indianapolis), though in all, tweets were detected in 46 of the 92 counties in the state. Scott County (population: 23,000), the location of the HIV outbreak, had only 4 relevant tweets, 3 of which were located in Scottsburg (population: 6,700). The 4 tweets from Scott County were primarily user reflections about the HIV outbreak, as listed here: (1) "The fact that our school is having an HIV meeting", (2) "Felt like an elementary schooler learning about my period become that hiv meeting, jeez", (3) "I feel like my opinion on this HIV outbreak would cause a lot of butthurt", and (4) "Scottsburg where the HIV be breakin out like a damn mad house. It sucks here. Come get me😊😊."

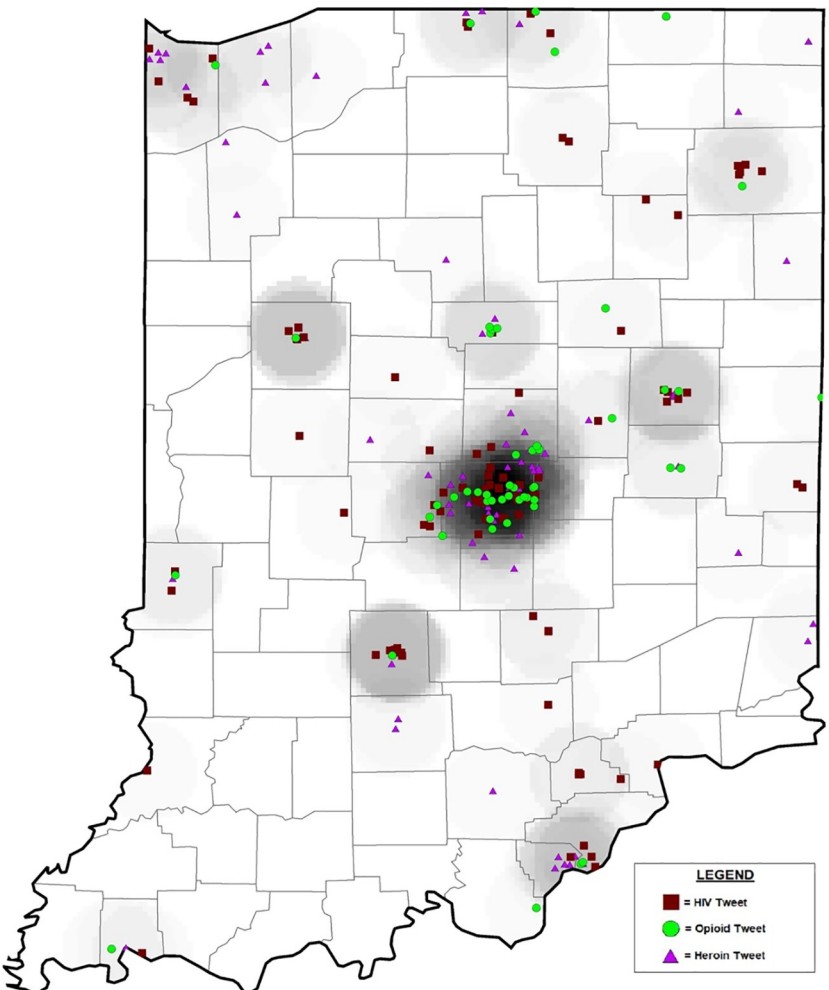

**Fig 3. Geospatial map of signal tweets in Indiana.**

In addition, we found tweets self-reporting heroin and prescription drug abuse-related behavior from the neighboring Clark County (population: 116,000), which borders Scott County, prior to the HIV outbreak. Such tweets included: "I just witnessed someone die of a heroin overdose" (October 2014); "Now thats it 2015 im finally going to kick my heroin addiction #newyearnewme" (January 2015); and "in for the meth and heroin" (January 2015). Collectively, the total volume of tweets that were detected in the outbreak area or adjacent to it were low, indicating that the majority of geolocated Twitter conversations about the outbreak or relevant to HIV-related risk behavior occurred outside of impacted communities.

## Discussion

This study conducted a retrospective analysis of tweets geolocated in Indiana during the immediate pre and post phase of the 2015 HIV outbreak. Specifically, the study used an unsupervised machine learning approach using NLP to detect messages related to user behavior associated with opioid abuse, heroin injection drug use, and HIV, critical elements of the risk environment that led to the 2015 Indiana outbreak. We did this in the hopes of finding specific "signal" of risk factors that may have preceded the outbreak. Overall, we found a relatively

small volume of Twitter conversations directly related to the Indiana outbreak and most conversations occurring in the post-outbreak period.

Based on our thematic analysis, the most common theme identified was opioid use, followed by heroin abuse. In Indiana, between 2005 and 2014, hospital admissions for heroin and prescription drug abuse increased 413% and 155% respectively, reflecting a growing problem of opioid-and heroin-related substance abuse behavior in the state [22]. In our dataset, we detected tweets self-reporting both heroin and opioid use, including examples of IDU and reports of overdose around Scott County. The demographic of IDU among people who inject drugs (PWIDs) and HIV in the USA has been primarily black, male, and urban, whereas the Indiana HIV outbreak demographic was among younger, white, and rural users [2]. This epidemiological profile is similar to current heroin users in the United States, potentially indicating that the demographic makeup of the state may have put users at higher risk for transition from opioid abuse to heroin injection drug use [2].

Our study design identified behavioral tweets reporting heroin injection drug use, opioid use, and HIV status, and also identified themes that are relevant to past and future prevention and health promotion efforts for HIV. Specifically, tweets we identified in our "prevention" theme illustrate that certain users expressed motivation to use prevention methods (e.g. safe sex practices, PrEP, HIV testing) and we also detected conversations circulating in impacted communities that may have both motivated or detracted those in the priority population from seeking proper HIV prevention options. Despite finding tweets related to HIV risk awareness, there were also tweets spreading misinformation about HIV transmission in afflicted communities, which could have potentially impacted vulnerable populations from seeking treatment interventions.

In addition, the political reaction and public sentiment tweets we detected provide insights on how public policies may have exacerbated the outbreak. These tweets focused on criticism towards then Indiana Governor Mike Pence and his refusal to authorize NEP despite recommendations from local public health leaders [19]. NEP has proven to be an effective harm reduction measure that can stop the spread of HIV among PWIDs and could have averted HIV cases in Indiana [22]. Other studies report the lack of appropriate harm-reduction services, including needle-exchange services and sterile syringe availability in rural Indiana as a major cause of the outbreak [2,19,23] Importantly, pre-outbreak, NEPs were not authorized, purchasing syringes without a prescription was illegal, and nonmedical use of syringes was a felony in the state [2,23] Tweets related to political reaction also focused on the defunding of Planned Parenthood, where in Scott County, the single Planned Parenthood clinic was the sole provider of HIV testing services prior to its closure in 2013.

In 2016, the U.S. Congress reversed a ban on using federal funds for syringe exchange programs, but many of the rural communities battling the opioid crisis are hesitant to implement comprehensive NEP and other harm reduction services, such as opioid agonist therapy [23] Hence, persistent barriers of stigma associated with addiction, drug abuse, and HIV status continues to limit access to preventive health and harm reduction services in vulnerable communities, including rural areas. There is also a lack of comprehensive national legislation that specifically addresses the intersection between the opioid, heroin, and HIV risks in the context of the current national opioid crisis [23] Collectively, obstacles remain to interrupting the dangerous transition from opioid abuse to heroin injection drug use and its associated HIV transmission risks, with new clusters of HIV cases among PWID also emerging in other US states [24,25].

## Limitations

Our study has certain limitations. First this study was limited to geolocated Twitter data and we did not purposely sample accounts or weight for user characteristics such as age, gender, or

other demographics. Hence, our study results are not generalizable and are not representative of the Twitter user community or Indiana residents. Additionally, the volume of geocoded tweets relative to non-geocoded tweets is relatively low (reported at approximately 1% of the full amount of Twitter content). Hence, it is likely that our filtering for geocoded tweets failed to capture the majority of conversations from users in Indiana, even though it would be equally hard to accurately identify if these tweets were from Indiana users without them being geocoded. Future studies should not only rely on geocoded data, but also examine user meta-data and mentions of locations in the text of the tweet or the user account information. We also did not interact with users of verify if they were actually engaged in opioid use disorder, injection drug use, or confirm their self-reporting of HIV risk behavior or status. Given this limitation, we cannot say with certainty that tweets detected regarding user statement of behavior are factual. However, validating the truthfulness of self-reported responses has similar limitations in other survey instruments.

## Conclusions

Infoveillance will likely play an important role in identifying future "signal" that an outbreak might occur due to increased conversations about behavioral risk factors as self-reported by social media users. This study provides an early glimpse of the potential of big data, unsupervised machine learning approaches and using infoveillance to inform data-driven HIV-IDU risk prevention and help develop evidence-based policymaking approaches.

## Author Contributions

**Conceptualization:** Mingxiang Cai, Neal Shah, Tim K. Mackey.

**Data curation:** Mingxiang Cai, Neal Shah, Jiawei Li, Wen-Hao Chen, Raphael E. Cuomo, Nick Obradovich, Tim K. Mackey.

**Formal analysis:** Mingxiang Cai, Neal Shah, Jiawei Li, Wen-Hao Chen, Raphael E. Cuomo, Tim K. Mackey.

**Funding acquisition:** Tim K. Mackey.

**Investigation:** Neal Shah, Wen-Hao Chen, Tim K. Mackey.

**Methodology:** Mingxiang Cai, Neal Shah, Jiawei Li, Wen-Hao Chen, Raphael E. Cuomo, Nick Obradovich, Tim K. Mackey.

**Project administration:** Neal Shah, Tim K. Mackey.

**Resources:** Tim K. Mackey.

**Software:** Mingxiang Cai.

**Supervision:** Tim K. Mackey.

**Validation:** Mingxiang Cai, Neal Shah, Jiawei Li, Wen-Hao Chen.

**Visualization:** Mingxiang Cai, Raphael E. Cuomo.

**Writing – original draft:** Mingxiang Cai, Neal Shah, Jiawei Li, Wen-Hao Chen, Raphael E. Cuomo, Tim K. Mackey.

**Writing – review & editing:** Mingxiang Cai, Neal Shah, Nick Obradovich, Tim K. Mackey.

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
