## [Decision Letter · Decision Letter 0]

10 Jun 2020

­Title: Identification and Characterization of Tweets related to the 2015 Indiana HIV Outbreak: A Retrospective Infoveillance Study

PONE-D-20-00556

We are pleased to inform you that your manuscript has been judged scientifically suitable for publication and will be formally accepted for publication once it complies with all outstanding technical requirements.

With kind regards,

Luigi Lavorgna

Academic Editor

PLOS ONE

In your Methods section, please include additional information about your dataset and ensure that you have included a statement specifying whether the collection method complied with the Twitter's terms and conditions. In addition, in the Methods section and online form, please include a statement about the studying being deemed exempt from IRB approval, and the need for consent being waived by the ethics committee.

Reviewers' comments:

Reviewer's Responses to Questions

**Comments to the Author**

1. Is the manuscript technically sound, and do the data support the conclusions?

Reviewer #1: Yes

Reviewer #2: Yes

2. Has the statistical analysis been performed appropriately and rigorously? 

Reviewer #1: Yes

Reviewer #2: Yes

3. Have the authors made all data underlying the findings in their manuscript fully available?

Reviewer #1: Yes

Reviewer #2: Yes

4. Is the manuscript presented in an intelligible fashion and written in standard English?

Reviewer #1: Yes

Reviewer #2: Yes

5. Review Comments to the Author

Reviewer #1: This manuscript provides an interesting approach to parlaying machine learning/artificial intelligence to analyze social media with the goal of having influence in the public heath sphere.

The writing is very clear, the methodology is described in clear detail and the authors do not overstep on their conclusions. This study does not suggest that for this particular case of the opioid overdose epidemic in Indiana in 2015 that the social media sphere would have been able to detect the outbreak. However, it does provide a description for tools that could possibly be of use in analogous public health emergencies.

I can think of multiple applications for the COVID-19 pandemic and social distancing behaviors which are being well documented on all social media platforms.

The powerful role of social media in people's lives is clear. AI is an important tool for detection of possible harms (self-harm, violence to others, disease outbreaks), and deserves in depth investigation of the sort presented in this (albeit negative) study.

Reviewer #2: This retrospective study offers an interesting and well designed point of view on an epidemiological infective outbreak in Indiana happened in 2015. The language used is correct and perfectly clear. Sample size appears wide enough to be significant for the purpose of this research and statistical analysis has been conducted appropriately. In my opinion, the authors have operated rigorously in the planning process, obtaining a wide range of results during the first selective part in the geographical media, the Twitter API. Unfortunately, some relevant data could have been lost due to privacy policy of single users. Then, thanks to the widely used and approved BTM followed by NLP process, all the results have been divided based on risk factors for HIV, IDU and heroin abuse, noticing a significant increase compared with precedent data. Correlation with the outbreak is potential and fortified by the NLP process that identified and removed news and non-generated user tweets. Furthermore, thanks to the analysis, the authors obtained data from the public intervention and its increase immediately after the outbreak. At the end a clear and exhaustive explanation on the limitations of this study offers new interesting starting-points for other studies. Could it be possible to evaluate similarly data from other social medias? Could it be possible to prevent future outbreaks monitoring the trend of some specific word patterns on the web making it a new alert for a ready public intervention? Those aspects could be evaluated in future studies.

6. PLOS authors have the option to publish the peer review history of their article (what does this mean?). If published, this will include your full peer review and any attached files.

Reviewer #1: No

Reviewer #2: No

---

## [Editor Report · Acceptance letter]

12 Aug 2020

PONE-D-20-00556 

Identification and Characterization of Tweets related to the 2015 Indiana HIV Outbreak: A Retrospective Infoveillance Study 

Dear Dr. Mackey:

I'm pleased to inform you that your manuscript has been deemed suitable for publication in PLOS ONE. Congratulations! Your manuscript is now with our production department. 

Kind regards, 

on behalf of

Dr. Luigi Lavorgna 

Academic Editor

PLOS ONE